# Development of PVA–Psyllium Husk Meshes via Emulsion Electrospinning: Preparation, Characterization, and Antibacterial Activity

**DOI:** 10.3390/polym14071490

**Published:** 2022-04-06

**Authors:** Fatma Nur Parın, Azeem Ullah, Ayşenur Yeşilyurt, Uğur Parın, Md. Kaiser Haider, Davood Kharaghani

**Affiliations:** 1Department of Polymer Materials Engineering, Faculty of Engineering and Natural Sciences, Bursa Technical University, Bursa 16310, Turkey; 2Nano Fusion Technology Research Group, Institute for Fiber Engineering (IFES), Interdisciplinary Cluster for Cutting Edge Research (ICCER), Shinshu University, Tokida 3-15-1, Ueda 386-8567, Nagano, Japan; 08tex101@gmail.com (A.U.); kaisershakil@yahoo.com (M.K.H.); 3Central Research Laboratory, Bursa Technical University, Bursa 16310, Turkey; aysenur.erdogan@btu.edu.tr; 4Department of Microbiology, Faculty of Veterinary Medicine, Aydın Adnan Menderes University, Aydın 09100, Turkey; uparin@adu.edu.tr; 5Department of Calcified Tissue Biology, Graduate School of Biomedical and Health Sciences, Hiroshima University, 1-2-3 Kasumi, Minami-Ku, Hiroshima 734-8553, Hiroshima, Japan

**Keywords:** PVA–psyllium husk electrospun meshes, D-limonene, modified microcrystalline cellulose, emulsion electrospinning, antibacterial activity

## Abstract

In this study, polyvinyl alcohol (PVA) and psyllium husk (PSH)/D-limonene electrospun meshes were produced by emulsion electrospinning for use as substrates to prevent the growth of bacteria. D-limonene and modified microcrystalline cellulose (mMCC) were preferred as antibacterial agents. SEM micrographs showed that PVA–PSH electrospun mesh with a 4% amount of D-limonene has the best average fiber distribution with 298.38 ± 62.8 nm. Moreover, the fiber morphology disrupts with the addition of 6% D-limonene. FT-IR spectroscopy was used to analyze the chemical structure between matrix–antibacterial agents (mMCC and D-limonene). Although there were some partial physical interactions in the FT-IR spectrum, no chemical reactions were seen between the matrixes and the antibacterial agents. The thermal properties of the meshes were determined using thermal gravimetric analysis (TGA). The thermal stability of the samples increased with the addition of mMCC. Further, the PVA–PSH–mMCC mesh had the highest value of contact angle (81° ± 4.05). The antibacterial activity of functional meshes against Gram (−) (*Escherichia coli, Pseudomonas aeruginosa*) and Gram (+) bacteria (*Staphylococcus aureus*) was specified based on a zone inhibition test. PPMD6 meshes had the highest antibacterial results with 21 mm, 16 mm, and 15 mm against Escherichia coli, Staphylococcus aureus, and Pseudomonas aeruginosa, respectively. While increasing the amount of D-limonene enhanced the antibacterial activity, it significantly decreased the amount of release in cases of excess D-limonene amount. Due to good fiber morphology, the highest D-limonene release value (83.1%) was observed in PPMD4 functional meshes. The developed functional meshes can be utilized as wound dressing material based on our data.

## 1. Introduction

Millions of people struggle with wounds and wound-related infections each year [1,2], and the global market for wound dressing products is estimated to exceed USD 15 billion by 2022 [2,3]. Conventional dressings, such as wet or dry medical gauze, must be changed frequently to protect wounds from contamination and microbial effects. However, these dressings may adhere to the wound surface, causing pain and soreness, and may damage healthy tissues [2]. Therefore, increased awareness of this fundamental aspect of wound care, or any trauma, is helping to develop newer strategies and approaches for therapeutic support, and innovative materials to improve the quality of human life. Injured skin can cause serious health problems such as loose thermal insulation, and issues with body fluid, electrolytes, and nutritional components in the human body [4,5,6,7].

A polymeric biomaterial is aimed to interact with biological systems in evaluating, repairing, restoring, or replacing any tissue, organ, or function of the body [8]. Polymers have great potential because of their chemical flexibility and a wide variety of physical and mechanical characteristics. Many bio-based polymers, such as gelatin, zein, chitosan, BSA, PCL, and PLA, have been employed in targeted drug delivery systems in various polymeric templates incorporating hydrogel, nanoparticle, nanocomposite, nanofilm, and nanofiber [9,10,11,12,13,14,15].

Among the many polymeric forms, nanofibers are sophisticated fibrous structures ranging in size from micrometers to nanometers [16]. The majority of electrospun nanofibers are limited to a densely packed two-dimensional (2D) mesh, which supports cellular proliferation only on its surface, restricting cell infiltration and proliferation through the nanofiber matrix [17]. In previous studies, a three-dimensional (3D) structure outperforms a typical two-dimensional (2D) nanofiber mesh, in cellular infiltration and proliferation. The three-dimensional structure simulates the extracellular matrix (ECM) and acts as a synthetic skeleton for cell proliferation, migration, and tissue repair [13]. The electrospun nanofibrous meshes display oriented in random 3D structures resemble the skin’s extracellular matrix (ECM). They have favorable wound healing properties [18]. The electrospun meshes have a high specific surface area, a high porosity, and linked pores, which create an ideal environment for cell growth and differentiation. The therapeutic agents and bioactive compounds (vitamins, growth factors, antimicrobials, antibacterials, anti-inflammatories) were introduced into nanofibrous materials by various approaches like core-shell, blend, emulsion electrospinning, and combining electrospraying and electrospinning [19,20,21,22].

Green electrospinning has grown in popularity in recent years due to its environmentally friendly, clean, and safe fabrication without toxic materials. As a result of this tendency, the emulsion electrospinning technique, which is the primary way of green electrospinning, has become attractive. The method of preparing polymer solutions is critical to emulsion electrospinning. An emulsion consists of two or more immiscible liquid phases, named continuous and disperse phases. Many studies on nanofibers with essential oils via emulsion electrospinning can be found in the literature [23,24]. Rieger et al. (2016) fabricated chitosan/polyethylene oxide/cinnamaldehyde nanofibers, and they observed relationships between the solution rheology and the final materials [25]. In another study, cabreuva essential oil was loaded into PVA–chitosan nanofibers produced by Lamarra et al. (2020) [26]. In their study, antibacterial activity against microorganisms (*Candida albicans, E. coli, S. aureus, and S. epidermidis*) and the release behavior of the resulting materials were evaluated. Mouro et al. (2019) developed composite electrospun fibers containing polycaprolactone (PCL)/polyvinyl alcohol (PVA)/chitosan (CS)/eugenol (EUG). They observed that the resultant nanofibers have a burst release of eugenol from the matrix for the first 8 h. Moreover, the antibacterial efficiency of the materials against *S. aureus* and *P. aeruginosa* bacteria has been found to be around 92% and 95%, respectively [27].

Emulsion electrospinning is receiving much interest for its capacity to successfully load both hydrophilic and hydrophobic substances and preserve these agents’ structural performance and bioactivity [18]. In the past few years, emulsion electrospinning has emerged as an effective alternative to conventional electrospinning. Moreover, with the development of new materials derived from renewable resources, a wide range of wall materials, including biopolymers (e.g., proteins, polysaccharides) and biocompatible polymers (e.g., polyvinyl alcohol (PVA), polyethylene oxide (PEO), and polycaprolactone (PCL), are suitable for emulsion electrospinning [28]. The electrospun emulsion fibers from these polymers have been intensively researched for biomedical and other areas, associated with the assessment of their biodegradability, biocompatibility, and low toxicity [29].

In comparison to conventional electrospinning techniques, the use of emulsion electrospinning is a promising alternative because it allows the encapsulation of lipophilic compounds using low-cost hydrophilic polymers, while avoiding the use of organic solvents, which are highly restricted in the biomedical area [30,31].

Aromatic oils obtained from plants have many natural antibacterial, antifungal, and antioxidant effects. There are many studies regarding the positive impact of these essential oils on healthy living as therapeutic components [4,31].

D-limonene is one of the most common terpenes in nature. It is a monocyclic monoterpene with a lemon-like odor and is the main ingredient in various citrus oils. It is widely used in perfumes, soaps, foods, and beverages due to its pleasant citrus fragrance additive. D-limonene (4-isopropenyl-1-methylcyclohexane) (C_10_H_16_) is the main component of orange peel (95%), lemon peel (75%), and tangerine and citrus peel (87%) oils [32,33,34]. The D-limonene essential oil can be utilized as a fragrance component in perfumes and creams, as an additive for food applications, and as an antibacterial and antifungal agent in medical applications [35,36,37]. The possible mechanism of antimicrobial activity is considered to be the disruption of the cell membrane by lipophilic compounds and the decrease in lipophilicity with the addition of the hydroxyl group [38,39].

A study by Fuenmayor et al. (2013) produced edible nanofibers containing D-limonene by the electrospinning method and suggested that it could be an antimicrobial material used in food applications [40]. In a similar study, D-limonene-loaded double-layered polymeric films were produced for food applications. It was determined that these active films showed antimicrobial activity against *E. coli, L. monocytogenes, S. aureus,* and *S. aureus P. aeruginosa* bacteria [41]. Lan et al. (2019) produced and characterized nanofibers from polyvinyl alcohol/D-limonene using an ultrasonic process [42]. These materials also exhibited antibacterial activity against *S. aureus* and *E. coli* bacteria.

There are numerous studies related to psyllium husk gel used to make fibers by wet spinning [6,7,43,44]. We know that studies concerning antibacterial nanofibers consisting of PVA–psyllium husk polyblends are not yet available in the literature. Based on the available literature, in this study, we aimed to develop new totally bio-based and antibacterial electrospun PVA–psyllium husk–D-limonene meshes via emulsion electrospinning. The morphological, chemical, and thermal analyses were examined. Finally, the antibacterial activity of the fibrous meshes against Gram (−) (*Escherichia coli, Pseudomonas aeruginosa*) and Gram (+) (*Staphylococcus aureus*) bacteria was assessed, as well as the release behavior of D-limonene in NaCl media, concerning the modified MCC (mMCC) and the varying D-limonene content (2%, 4%, and 6%, *v/v*).

## 2. Materials and Methods

### 2.1. Materials

PVA (purity 87.8%, Mw~30.000 g/mol) was supplied from ZAG Industrial Chemicals (Istanbul, Turkey). Psyllium husk powder was purchased from Naturebyme (İstanbul, Turkey). Microcrystalline cellulose (MCC) and D-limonene (%99 purity) were also purchased from Kimbiotek Chemical Substances Industry Trade Inc. (İstanbul, Turkey). The (3-chloropropyl) triethoxysilane (CPTES) and glacial acetic acid (CH_3_COOH, 99%) were bought from Sigma-Aldrich (St Louis, MO, USA). Merck supplied Trypton Soy Agar (TSA, 105458) (Kirkland, QC, Canada). Distilled water was used in all experiments, and all materials were utilized as received.

### 2.2. Preparation of Modified Microcrystalline Cellulose

An amount of 5 mL of (3-chloropropyl) triethoxysilane (CPTES) was dispersed in 100 mL of distilled water, and then glacial acetic acid was added until the pH reached 4 (acidic condition) (Figure 1). Next, MCC was introduced into the mixture and blended for 2 h at room temperature. The mixture was centrifuged at 11,000 rpm for 20 min, according to the literature [45]. The remaining unreacted particles were poured and washed once with water. Finally, the bulk material was put in an oven at 40 °C for 24 h (Figure 2).

### 2.3. Fabrication of Polysaccharide Nanofibers

PVA granules were dissolved in distilled water to form 10% (*w*/*v*) PVA solutions by stirring at 90 °C for 3 h. Meanwhile, psyllium husk powders were dispersed in 10 mL distilled water using an ultrasonic homogenizer (Bandelin/Sonoplus HD3200) to obtain a 1% polymer solution. Afterward, PVA and psyllium husk polymer solutions were mixed to prepare blend solutions (4/1, *v*/*v*). The modified MCC (0.012 g) was added to the blend polymer solution. Then, D-limonene was introduced to be 2%, 4%, and 6% of the total polymer solutions. The nanofibers were electrospun at a high voltage of 25 kV and a flow rate of 1.25 mL/h, with a spinning drum at 250 rpm (Figure 3). The drum was coated with aluminum foil, and the distance between the tip and the collector was 88 mm. These samples were coded in Table 1.

### 2.4. Characterization

The morphology of the resulting nanofibers was investigated using a Carl Zeiss/Gemini 300 Scanning Electron Microscope (SEM) (ZEISS Ltd., Aalen, Germany). At 5 kV, microstructure analyses were performed on the surface area of these samples. All the samples were coated with gold–palladium before the analysis. SEM images at 10.000, 5.000, and 1.000 magnifications were acquired. Then, Image J (Software version 1.520) was used to measure the pore size diameters, by selecting the diameters of 100 pores at random for each sample.

The chemical bonding of fabricated nanofibers was determined using a Fourier Transform Infrared (FT-IR) spectrometer with Smart Orbit Diamond ATR (Attenuated Total Replication) adapter (Thermo Fisher Nicolet IS50, Waltham, MA, USA). The measurements were taken in the wavelength range of 4000–500 cm^−1^, with 16 scans recorded at 4 cm^−1^ resolutions.

Thermal stability of the nanofibers, MCC, and mMCC was carried out using a TA/SDT650 TGA (USA), in a nitrogen atmosphere with a heating rate of 10 °C min^−1^ over temperatures ranging from room temperatures to 600 °C, and in an oxygen atmosphere with a heating rate of 10 °C min^−1^ over a temperature range of 600–900 °C. Similarly, TGA analysis of D-limonene was applied in a nitrogen atmosphere with a heating rate of 10 °C min^−1^ over temperatures ranging from room temperatures to 600 °C.

The hydrophilic behavior of the obtained electrospun meshes has been investigated for contact angles. An Attention Thetaflex optical tensiometer (Biolin Scientific, Gothenburg, Sweden) was used, and 5 µL distilled water samples were dropped by 2 cm × 2 cm.

### 2.5. In Vitro Release Study

In vitro releasing tests (IVRT) were performed to evaluate the D-limonene release behavior of resultant nanofibers, based on the total immersion method [46]. Each sample (25 cm^2^) was put in sterilized bottles with 100 mL of NaCl (0.8%) solutions. After that, the samples were placed in a shaking incubator at 37 °C with stirring at 120 rpm. An amount of 3.5 mL of the samples were withdrawn with NaCl solutions at the specified time intervals. The corresponding absorbance value was recorded using a UV spectrophotometer (Scinco/NEOYSY 2000) set to max = 237 nm (the typical peak of D-limonene). The D-limonene concentration was determined using the measured calibration curve, which was constructed using a known amount of D-limonene solution in NaCl. The calibration curve was calculated as Y = 0.3689X + (0.2078) (R^2^ = 0.96), where X is the D-limonene concentration (mL/L) and Y is the solution absorbance at 237 nm. The cumulative releases (%) were determined using the following equation:(1)$$\mathrm{Cumulative}\;\mathrm{release}\;(\%)=\sum_{t_{0}}^{t}\frac{M_{t}}{M_{0}}\times 100$$
where *M_t_* is the total amount of D-limonene released at usual time intervals, *t* and *M*_0_ are the beginning amounts of D-limonene in the electrospun meshes.

Encapsulation efficiency has also been calculated according to the following equation:(2)$$\mathrm{Encapsulation}\;\mathrm{efficiency}\;(\%)=\frac{\mathrm{Amount}\;\mathrm{of}\;\max\;D-\mathrm{limonene}\;\mathrm{release}}{\mathrm{Theorical}\;\mathrm{amount}\;\mathrm{of}\;\mathrm{total}\;D-\mathrm{limonene}\;\mathrm{added}\;\mathrm{in}\;\mathrm{meshes}\;}\;\times \;100$$

### 2.6. Antimicrobial Efficiency

The antimicrobial efficacy of electrospun meshes was evaluated with the standard strains of Escherichia coli ATCC^®^ 25922 and Staphylococcus aureus ATCC^®^ 25923. Trypton Soy Agar (Merck Millipore™ 105458, Burlington, MA, USA) was used to grow the lyophilized bacterial strains. The inoculated agar plates were incubated for 24 h (37 °C) in aerobic conditions. Bacterial suspensions were adjusted to 0.5 McFarland (1 × 10^8^ CFU/mL) turbidity in an isotonic sodium chloride solution.

Antibacterial properties were specified qualitatively by the disk diffusion method. The bacterial suspensions were inoculated across the surface of Mueller Hinton Agar (Merck Millipore™ 103872) with a volume of 100 µL. The sponges to be examined were placed on the plates in contact with the agar surface after drying. The plates were incubated for 24 h. The diameter of the inhibition zones around the material inserted for each sponge was qualitatively measured and analyzed at the end of the incubation period.

## 3. Results and Discussion

### 3.1. Microstructure and Morphological Analysis

As can be seen in Figure 4(A1–B2), electrospun PVA–PSH meshes are thinner than PVA meshes, and the addition of PSH partially destroyed the homogeneity in the structure. Additionally, the existence of modified MCC in polyblend PVA–PSH meshes (Figure 4(C1)) led to a straighter structure, some deterioration of structural integrity, and an increased average fiber diameter with 340.1 ± 104 nm (Figure 4(C2)). The presence of only D-limonene in the fiber without modified MCC (PPD samples) also showed that it could mix well with the matrix due to the beta-cyclodextrin (Figure 4(D1)). It is considered that solid modified MCC may cause agglomeration within the fiber.

The increased amount of D-limonene in polyblend meshes caused a decrease in solution viscosity and a decrease in fiber thickness (Figure 4(E1–G2)). Generally, as the viscosity of the spinning solution increases, fiber diameter increases [47]. Further, rapid evaporation of essential oils can also cause this issue. That means the fibers in the collector can be obtained thinner owing to the partial removal of the other essential oil, as reported in previous studies [24,48]. For the mesh with the highest D-limonene content (PPMD6), the morphology of the composite electrospun meshes did change considerably due to the bead structures observed. The thinning of the fiber diameter can also increase surface area, which can speed up the wound healing process. PPMD2 meshes have the largest and smallest fiber diameters among the electrospun meshes, with 131 nm and 778 nm values. Nevertheless, the best average fiber distribution was in the PPMD4 meshes (Figure 4(F1,F2)).

### 3.2. FT-IR Analysis

Figure 5 exhibited the FT-IR spectra of the D-limonene, mMCC, pure PVA, PVA–PSH, and PVA–PSH–mMCC nanofibers, and evaluated physical interactions between the polyblends (PVA–PSH) and D-limonene.

The FT-IR spectra of D-limonene displayed characteristic peaks at 796 and 885 cm^−1^ according to C–H bending, 1644 cm^−1^ due to C=C stretching, and 2833, 2916, and 2964 cm^−1^ due to C–H stretching [42]. In the mMCC, the wide band formed in the 3100–3600 cm^−1^ range, related to –OH stretching vibration, provides significant information on hydrogen bonding. At 1428 cm^−1^, there is also an FT-IR absorption band ascribed to a symmetric –CH_2_ bending vibration. This band is commonly referred to as the “crystallinity band”. Peaks at 1054 and 1314 cm^−1^ correspond to the C–O–C pyranose ring skeletal vibration, respectively [49]. The FT-IR characteristic peaks of pure PVA nanofiber (PP0), broadband at 3296 cm^−1^, are assigned to the –OH stretching vibrations. The peaks at 2949 and 2914 cm^−1^ are related to asymmetric and symmetric –CH_2_ stretching vibration, respectively [33]. A band at 1427 cm^−1^ is assigned to the C–H bending vibration. The peak at 1243 cm^−1^ is due to the O–H bending vibration. The spectrum of PVA–PSH meshes is similar to PVA samples. However, a decrease in peak intensity has occurred. PVA–PSH samples have broadband at 3305 cm^−1^ (–OH stretching vibrations), 2938, and 2907 cm^−1^ (asymmetric and symmetric –CH_2_ stretching vibrations), respectively. MCC could not be detected in PPM and other samples due to adding a small MCC amount. As D-limonene was incorporated into the polyblend composition (PPD), relatively slight peak shifts in the spectrum were observed. A remarkable difference in peak trends has occurred at 1081 and 1025 cm^−1^. It is considered that this is due to the predominance of the characteristic peaks of D-limonene in these bands, and that the polyblend matrix overlaps the other distinct peaks of D-limonene. The difference in the bands (1080–780 cm^−1^) is clearly seen in the PPMD6 samples.

### 3.3. Thermal Stability Test

TGA has been used to investigate D-limonene, MCC, mMCC, and all the nanofibers’ thermal stability, and the TGA thermograms are presented in Figure 6 and Figure 7. All electrospun samples’ weight loss in the range of about 30–100 °C (first steps) was due to moisture loss from the samples. For the PP0 sample, three degradation steps have occurred. The second step (200–400 °C) of weight loss was correlated with the degradation of polymer side chains in the sample. The breakdown of C–C bonding from PVA occurred in the last step (600–650 °C).

Thermal degradation of a polysaccharide commonly involves four steps, each corresponding to the specific degradation behavior of the polysaccharide [50]. In PP1 samples, by adding the PSH to the polymer matrix, it is apparent in the first step that time is needed for the moisture to be lost from the samples. Moreover, the second stage (200–400 °C) shows a decrease in weight attributed to the backbone breakage of both PVA and PSH (C–O and C–C bonds in the ring units). In the third step (400–550 °C), molecules such as CO, CO_2_, and H_2_O have formed due to the chain breaks resulting from the depolymerization. In the final step (580–650 °C), polyblend chains are degraded completely, and aromatic compounds and graphitic carbon structures are formed. For PPM meshes, modified MCC has slightly changed degradation behavior. Although degradation occurs at almost the same temperature ranges (200–400 °C), the thermogram is broadened in the second step. PPD meshes also exhibited three stages of weight loss after moisture loss. However, in the last step (after around 600 °C), the residues formed in the second and third steps, and the cyclic group in the D-limonene in the structure and the opening of the cyclodextrin rings, and then a chemical progression with the loss of hydroxyl groups and glucosidic structure, and the formation of unsaturation, carbonyl groups and aromatic structures [51]. For PPMD2, PPMD4, and PPMD6 samples, peak splits have occurred in the first step as the polyblend structure is more complex. Therefore, the thermogram is complicated in the pyrolysis step.

### 3.4. In Vitro Release Study

The release behavior of polyblend–D-limonene meshes was evaluated by UV-Vis spectroscopy for 8 h (in NaCl media). Figure 8a,b presents the cumulative D-limonene release amount in NaCl media from the PVA–PSH polyblend meshes. The total D-limonene releases for PPD, PPMD2, PPMD4, and PPMD6 are 0.27, 0.75, 0.95, and 0.55 ppm, respectively, based on the release behavior. Moreover, the cumulative release of D-limonene from the meshes (PPD, PPMD2, PPMD4, and PPMD6) are 47.3%, 61.4%, 83.1% and 54.3%, respectively. At the initial analysis stage, all specimens displayed a significant fast release characteristic. Further, except for the PPMD4 meshes, all the specimens released more than half of the D-limonene amount. This sample also revealed the highest D-limonene release. This case can be associated with the increase in the surface area and the increase in the amount of D-limonene release because of the decreasing of average fiber diameter [22,52].

Meanwhile, excess D-limonene deteriorated the fiber morphology, caused beaded fiber formation, and reduced the release amount. These phenomena are in line with the literature. Çallıoğlu and Güler (2020) fabricated PVP/gelatin/lavandula essential oil nanofibers [24]. They asserted that a certain concentration of lavender causes beads in their nanofibers and their morphological deformation begins. It has also been found that if the amount of Zataria Multiflora (ZM) essential oil is >10% (*v*/*v*), the continuous form of the polymer solution deforms and therefore no nanofibers are produced [53]. Many factors contribute to the fast release of D-limonene from polyblend meshes, including the hydrophilic character of the matrixes (PVA and PSH) and the large surface area of the fibers, which enables an increase in wettability [54,55]. When comparing PPD and PPMD2 meshes, it appears that introducing mMCC to electrospun meshes enhances both the D-limonene release amount and controlled release. The presence of mMCC in the meshes improved the hydrophilicity of the samples, and this result is in good accordance with contact angle results. In brief, electrospun PVA–PSH meshes including D-limonene have a fast release at first, followed by a gradual release behavior. Due to the dissolution of the entire meshes in the release medium, the encapsulation efficiency is also equal to the maximum amount of release.

### 3.5. Contact Angle Analysis

Wettability can be the initial evidence for forecasting the biocompatibility of non-biological materials, since the generation of a protein layer in the first interaction with biological systems is the first stage and first indication of the material’s potential compatibility [56]. Materials with contact angles of 0–30°, 30–90°, and higher than 90° are defined as hydrophilic, semi-hydrophilic, and hydrophobic, respectively [57,58].

Wound care is the continued treatment of a wound by maintaining an effective environment for regeneration, through both direct and indirect approaches, and the prevention of skin destruction [59]. A moist microenvironment has been shown to aid wound healing by reducing dehydration, promoting angiogenesis and collagen formation, and increasing dead tissue and fibrin breakdown. This enhances the wound’s appearance while reducing discomfort [60].

With the addition of D-limonene oil, the water contact angle decreased (Figure 9). This may be due to the fact that β-cyclodextrin containing both hydrophilic and hydrophobic properties [56]. Therefore, the contact angle value for pure PVA–PSH and PVA–PSH–beta-cyclodextrin was (68.56° ± 3.4°) and (48° ± 2.4°), respectively.

PP0 meshes showed super hydrophilic properties. Because of the hydrophilic nature of both polymers, electrospun PVA–psyllium husk polyblend meshes have a low water contact angle (68.56°), which is ideal for usage in wounds applications. It is seen that the meshes containing modified MCC have the highest value with 81° ± 4.05. Although the hydrolysis procedure used in the CPTES treatment partially reduced the hydrophobicity of the MCC, the hydrophobic nature of the MCC predominated, and there was a significant difference in the wettability of the pure PP1 sample.

D-Limonene is a highly hydrophobic molecule that is almost insoluble in water [61]. In general, the contact angle decreased with the addition of D-limonene to the mesh structures due to the existence of the β-cyclodextrin substance. However, as the concentration of D-limonene increased, the structure tended to become linearly more hydrophobic (PPMD2 30.1, PPMD4 42.12, PPMD6 45.31).

### 3.6. Antibacterial Assays

The disk diffusion procedure was performed to evaluate the antibacterial activities of D-limonene-loaded meshes and D-limonene-free meshes, against three kinds of microbial species (*E. coli, S. aureus, and P. aeruginosa*) bacteria (Table 2 and Figure 10).

The antibacterial activity of D-limonene as an antimicrobial agent has been reported by other authors [62,63]. D-limonene has been shown to strongly inhibit both Gram-negative and Gram-positive bacteria, as well as fungal activity [64]. Furthermore, several studies have shown that D-limonene may effectively prevent the development of spoilage bacteria such as *Aspergillus niger*, *Pseudomonas aeruginosa*, *Staphylococcus aureus*, and *Escherichia coli* [65,66]. Gram (−) bacteria are often more resistant to antibacterial agents than Gram (+) bacteria. Essential oils perform as antimicrobial agents by destroying the phospholipids found in cell membranes, causing increased permeability and cytoplasm leakage, or interacting with enzymes found on the cell wall. Thus, as D-limonene concentration increased in polyblend meshes, the meshes indicated increased antibacterial efficacy for all bacteria. The peptidoglycan cell wall is enclosed by an exterior lipopolysaccharide wall in Gram (−) bacteria. The decreased antibacterial efficacy of essential oils against Gram (−) bacteria is due to cell wall lipopolysaccharides limiting ingredient transport.

Moreover, contrary to the literature, the PPMD6 meshes showed the highest antibacterial effect against *E. coli*, with 21 mm zone inhibition. According to the literature, Lan et al. (2019) found the antibacterial activity grew as D-limonene increased. Nevertheless, a further increase in D-limonene reduces the antibacterial effect by changing the fiber morphology [42]. Recently, Estevez Areco et al. (2022) studied the antibacterial activity of PVA–R-limonene and PVA–beta-cyclodextrin–R-limonene nanofibers [67]. The PVA–R-limonene and PVA–beta-cyclodextrin–R-limonene nanofibers have almost the same antibacterial activity as *E. coli* growth. This is due to the toxicity effect of cyclodextrin on *E. coli*. Moreover, the low antibacterial activity of nanofibers was attributed to the low use of R-limonene (2.6 mL in total solution). In another study, Aytac et al. (2016) confirmed the potent antibacterial activity of cyclodextrin nanofibers, including D-limonene, against *E. coli and S. aureus* bacteria at ranges from 70–97% value [68].

Although the cellulose polymer does not have antibacterial properties, its high biocompatibility, aspect ratio, and functionality degree have resulted in the antimicrobial use of cellulose nanocomposites with several antimicrobial agents [69].

## 4. Conclusions

PVA–PSH electrospun meshes containing both D-limonene and mMCC were efficiently produced by emulsion electrospinning, and D-limonene was used to model an antibacterial agent. MCC was treated with a silane coupling agent, which is known as (3-chloropropyl) triethoxysilane (CPTES), to obtain mMCC. The microstructural analysis indicated that an excessive increase in the amount of D-limonene caused beaded fiber formation. Moreover, the average fiber diameter of meshes decreased from 456.53 ± 56.9 nm (for PP1) to 274.22 ± 68.25 nm (PPMD6), due to the increase in D-limonene. In FT-IR spectroscopy, the characteristic peaks of the matrix overlapped the distinct peaks of D-limonene. The characteristic peaks of D-limonene were appointed slightly in the 1000–1200 cm^−1^ band. Further, only the addition of MCC increased the thermal stability, based on thermal analysis. It was reported that the increase of D-limonene increases the contact angle, but the contact angle of all D-limonene-containing samples may be lower due to β-cyclodextrin compared to the neat sample (PP1). The modified MCC added to polyblend meshes increased the release of D-limonene and the antibacterial activity of the resulting meshes. According to the findings, D-limonene revealed more antibacterial activity against Gram-negative bacteria (*E. coli*) than Gram-positive bacteria (*S. aeurus*), highlighting that electrospun meshes are suitable for carrying D-limonene, effectively increasing its antimicrobial activity. It supports the evidence that D-limonene continues in its therapeutically active form once loaded into electrospun fibers. We hope that our results will encourage people to recognize the importance of traditional natural molecules, such as D-limonene, for their potential therapeutic advantages, and develop the pharmaceutical and cosmetic fields for efficient protection and release of such natural molecules. Future research should investigate bioactive agents’ release behavior and associated activities (combining other antibacterial molecules with D-limonene), when loaded in electrospun fibers made from diverse polymer combinations (PSH and another matrix). The formulations are likely to be more successful when tested in appropriate in vivo models. In summary, the study demonstrated that D-limonene might support as an inhibitor of bacteria. Therefore, the obtained new polyblend meshes can be candidates for wound healing applications.

## Figures and Tables

**Figure 1 polymers-14-01490-f001:**
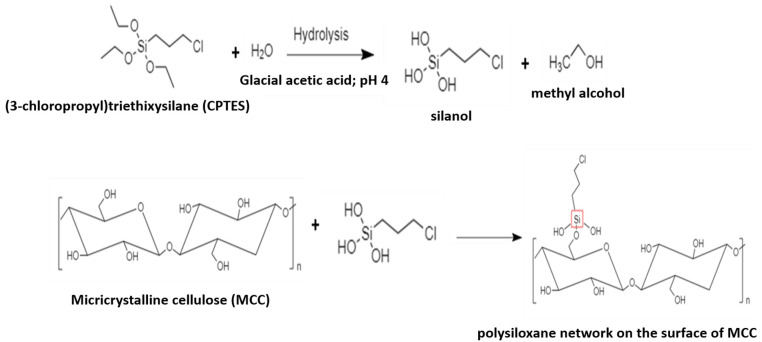
Reaction process of the modified microcrystalline cellulose.

**Figure 2 polymers-14-01490-f002:**
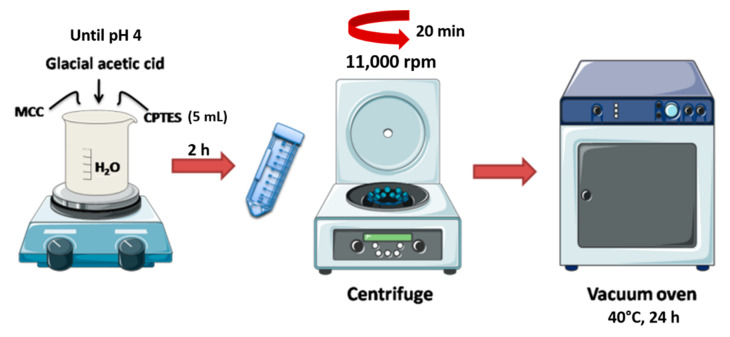
Schematic illustration of modified MCC (mMCC) modification.

**Figure 3 polymers-14-01490-f003:**
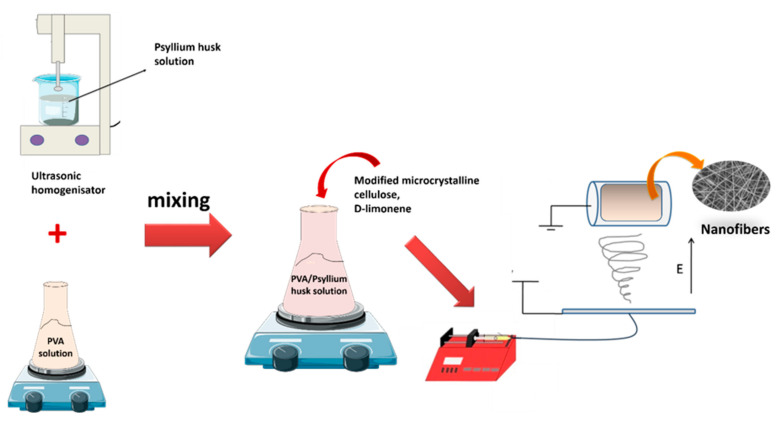
Schematic illustration of modified MCC modification.

**Figure 4 polymers-14-01490-f004:**
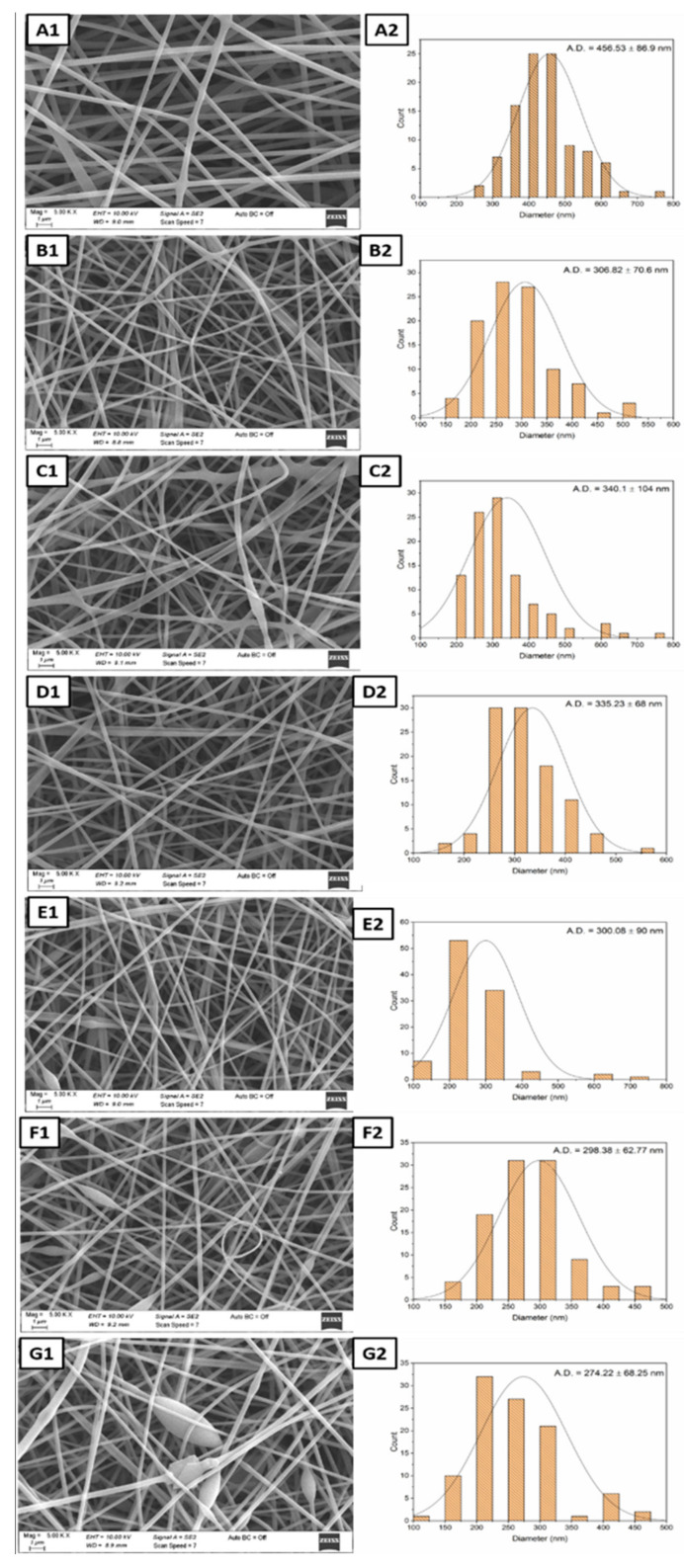
SEM micrographs and average diameters of the nanofibers (**A1**,**A2**) PP0; (**B1**,**B2**) PP1; (**C1**,**C2**) PPM; (**D1**,**D2**) PPD; (**E1**,**E2**) PPMD2; (**F1**,**F2**) PPMD4; and (**G1**,**G2**) PPMD6.

**Figure 5 polymers-14-01490-f005:**
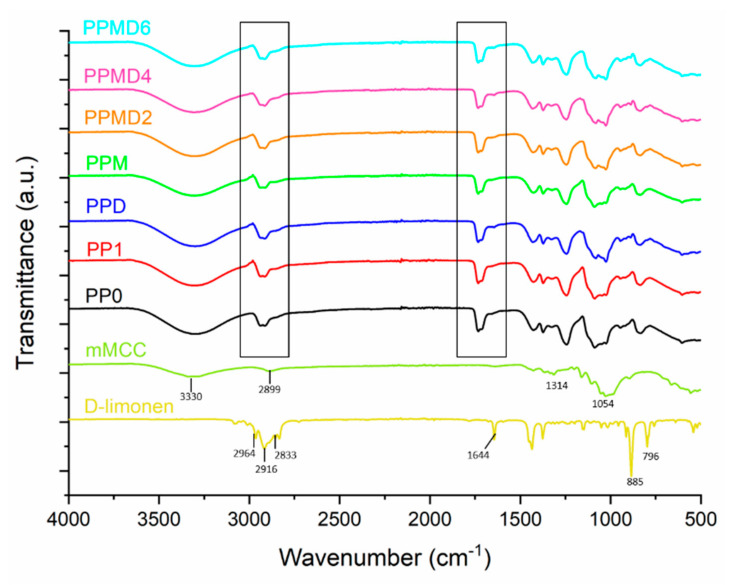
FT-IR spectra of the polyblend meshes.

**Figure 6 polymers-14-01490-f006:**
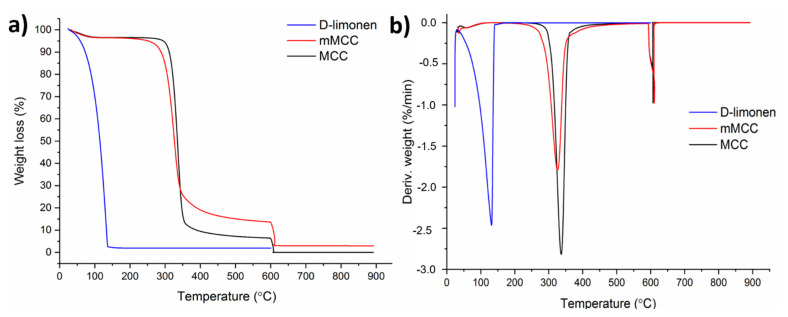
(**a**) TGA and (**b**) DTG thermograms of pristine MCC, modified MCC, and D-limonene.

**Figure 7 polymers-14-01490-f007:**
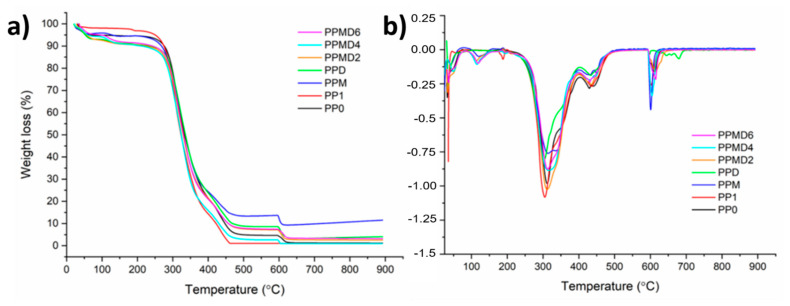
(**a**) TGA and (**b**) DTG thermograms of all electrospun meshes.

**Figure 8 polymers-14-01490-f008:**
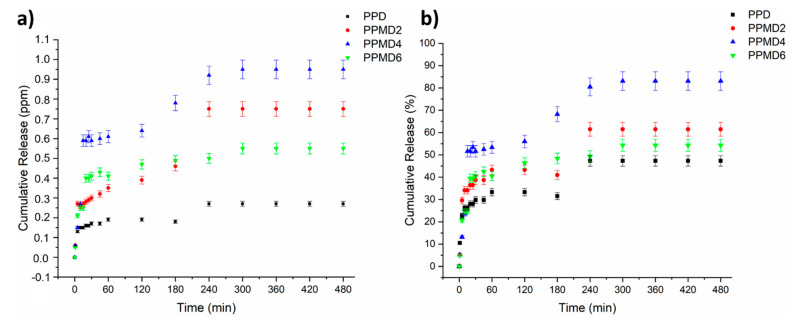
Cumulative D-limonene (**a**) (PPM) and (**b**) % release from the PVA–PSH electrospun meshes.

**Figure 9 polymers-14-01490-f009:**
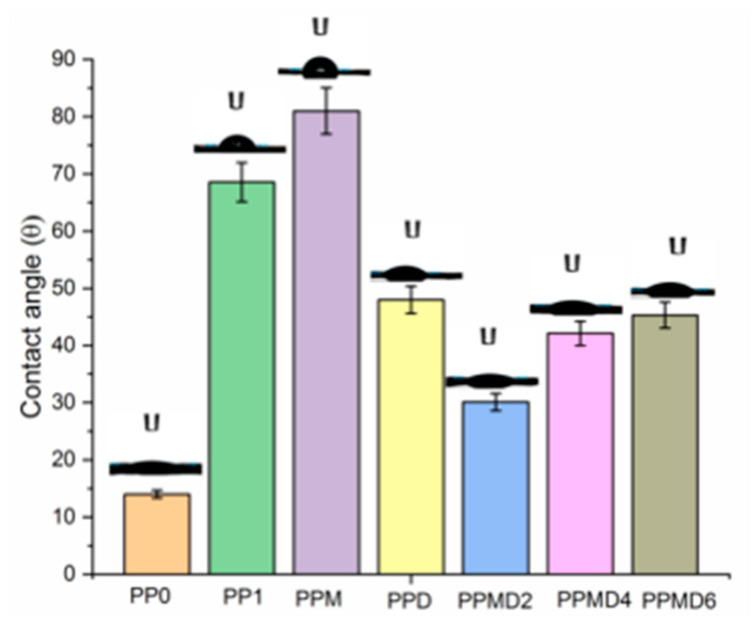
Contact angle values of all electrospun meshes.

**Figure 10 polymers-14-01490-f010:**
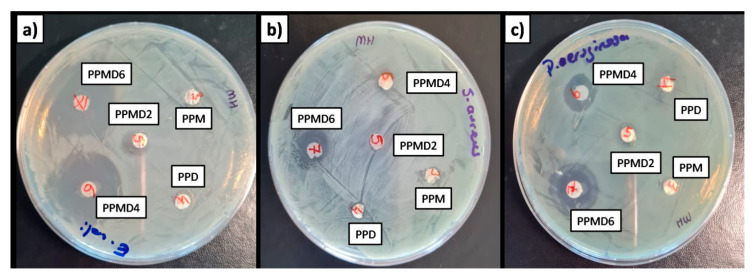
Zone inhibition of electrospun meshes containing D-limonene and mMCC against (**a**) *E. coli*, (**b**) *S. aureus*, and (**c**) *P. aeruginosa*.

**Table 1 polymers-14-01490-t001:** The compositions of electrospun meshes.

Sample ID	PVA (10%)	Psyllium Husk (1%)	mMCC Amount (%)	D-Limonene Amount (%)
**PP0**	4	-	-	-
**PP1**	4	1	-	-
**PPD**	4	1	-	2
**PPM**	4	1	0.012	-
**PPMD2**	4	1	0.012	2
**PPMD4**	4	1	0.012	4
**PPMD6**	4	1	0.012	6

**Table 2 polymers-14-01490-t002:** Inhibition zone of the electrospun meshes.

Bacteria	Zone Inhibition (mm)						
	PP0	PP1	PPM	PPD	PPMD2	PPMD4	PPMD6
*E. coli*	-	-	8	9	12	18	21
*S. aureus*	-	-	8	9	11	13	16
*P. aeruginosa*	-	-	8	9	11	13	15

## Data Availability

Data are available upon request from the corresponding author.

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
