# Peer review of "Development of PVA–Psyllium Husk Meshes via Emulsion Electrospinning: Preparation, Characterization, and Antibacterial Activity"

_polymers, 2022, doi:10.3390/polym14071490_

Round 1
Reviewer 1 Report
The authors have prepared a research article entitled “Development of PVA/Psyllium husk meshes via emulsion electrospinning: Preparation, characterization, and antibacterial activity”. The article has some interesting results and the authors have made considerable attention to preparing it. However, some issues need to be clarified before further consideration. Thus, the reviewer recommends this work can be published after a major review.
The reviewer has the following comments
-
- Rewrite the abstract. Remove very generic statements and focus directly on the original findings of the current research.
- The following information should be rephrased
“The thermal stability of the meshes was enhanced by the mMCC and, the PVA/PSH/mMCC mesh had the highest value of the contact angle (81° ± 4.05). The antibacterial activity of functional meshes against Gram (-) (Escherichia coli, Pseudomonas aeruginosa) and Gram (+) bacteria (Staphylococcus aureus) increased with an increase in D-limonene concentration. While increasing the amount of D-limonene enhanced the antibacterial activity, it significantly decreased the amount of release in case of excess D-limonene amount”.
- The statistical analysis section should be added to the revised manuscript.
- Names of each chemical structure should be reported in Figure 1.
- Figures 1 to 3 are not cited in the manuscript.
- Pristine MCC and D-limonene should be also included in the FTIR spectra to show the chemical or physical interactions. Peak frequency values should be also denoted in the FTIR spectra.
- Pristine MCC and D-limonene should be also included in Figure 6 to compare the thermal behavior of the materials.
- The following statement was made by the authors and which should be proved by some evidence, i.e., SEM of the nanofibers with an excess of D-limonene should be reported.
“Meanwhile, excess D-limonene deteriorated the fiber morphology, caused beaded fiber formation, and reduced the release amount.”
- SEM images of fibers after drug-loading and drug-releasing should be reported.
- Did the authors do the crosslinking of PVA and its composites fibers? Because without the crosslinking process the fibers immediately dissolved in the aqueous solution/buffer. Therefore, it is difficult to measure drug release.
- The authors should report the pH sensitivity of pristine and composite nanofibers. Because they behave differently under a wide range of pHs. Thus, studying the drug release under physiological and other pH solutions is recommended. The authors may follow the following articles as model procedures for pH-responsive systems and quote them in the revised manuscript. https://doi.org/10.1039/D1EN00354B, https://doi.org/10.1016/j.msec.2020.111698, https://doi.org/10.1016/j.msec.2020.110928
- The mechanical properties of nanofibers are the key characteristics for advanced in vivo drug delivery and tissue engineering applications. The authors should be reported the tensile data in the revised manuscript.
- The introduction section is insufficient and should be revised entirely so that the rn identify the scientific problems solved by this research. Several biocompatible systems have a wide range of biomaterials, including hydrogels, nanoparticles, nanofilms, nanocomposites, etc., but why did the authors select only nanofibers? The authors should emphasize why the nanofibers are familiar, or favor compared to other systems using the following articles. Moreover, the information on biomaterials (BSA, Gelatin, Zein, PCL, PLA, chitosan, UHMWPE, etc.) should be explored in the introduction with recent references, Thus, the following articles should be quoted in the introduction and other sections.
Chitosan (https://doi.org/10.1016/j.colsurfb.2021.111819), BSA (https://doi.org/10.1016/j.msec.2020.111698), Gelatin; (https://doi.org/10.3390/ph14040291, https://doi.org/10.1016/j.jmbbm.2020.103696), Zein (https://doi.org/10.3390/pharmaceutics11120621). https://doi.org/10.1016/j.msec.2020.110928. https://doi.org/10.1039/D1EN00354B, https://doi.org/10.3390/pharmaceutics12121208, https://doi.org/10.1016/j.jmbbm.2021.104554
It would be more realistic to cover such kind of research work in the current manuscript. Which will enrich the quality of the current manuscript as well as inquisitiveness to the readers.
Author Response
Manuscript Number: polymers-1647211
Point to point revision
Manuscript Title: Development of PVA/Psyllium husk meshes via emulsion electrospinning: Preparation, characterization, and antibacterial activity.
Dear Editor,
We thank the reviewers for the evaluations, suggestions and constructive comments on the development of our manuscript. Changes and some additions were suggested by referees in manuscript have been made. The given answers are presented below one by one.
Reviewer 1
The reviewer has the following comments
Comments and Suggestions for Authors
The authors have prepared a research article entitled “Development of PVA/Psyllium husk meshes via emulsion electrospinning: Preparation, characterization, and antibacterial activity”. The article has some interesting results and the authors have made considerable attention to preparing it. However, some issues need to be clarified before further consideration. Thus, the reviewer recommends this work can be published after a major review.
The reviewer has the following comments
- Rewrite the abstract. Remove very generic statements and focus directly on the original findings of the current research.
- Abstract has been rewritten again (in red).
- The following information should be rephrased
“The thermal stability of the meshes was enhanced by the mMCC and, the PVA/PSH/mMCC mesh had the highest value of the contact angle (81° ± 4.05). The antibacterial activity of functional meshes against Gram (-) (Escherichia coli, Pseudomonas aeruginosa) and Gram (+) bacteria (Staphylococcus aureus) increased with an increase in D-limonene concentration. While increasing the amount of D-limonene enhanced the antibacterial activity, it significantly decreased the amount of release in case of excess D-limonene amount”.
- The thermal stability of the meshes was improved by the addition of mMCC, and the PVA/PSH/mMCC meshes showed the highest contact angle value (81°±4.05°). Antibacterial activity of functional meshes against Gram (-) (Escherichia coli, Pseudomonas aeruginosa) and Gram (+) bacteria (Staphylococcus aureus) increased due to the concentration of D-limonene. The amount of D-limonene in the meshes increased antibacterial activity, while an excess amount of D-limonene significantly decreased in-vitro release values.
- The statistical analysis section should be added to the revised manuscript.
- Dear reviewer, thank you for your attention. However, we cannot do the statistical analysis in the study.
- Names of each chemical structure should be reported in Figure 1.
- The names of chemical structures have been added in Figure 1.
- Figures 1 to 3 are not cited in the manuscript.
- Figure 1 and 3 have been cited in the manuscript.
- Pristine MCC and D-limonene should be also included in the FTIR spectra to show the chemical or physical interactions. Peak frequency values should be also denoted in the FTIR spectra.
- Pristine MCC and D-limonene have been added to FTIR spectra.
- Pristine MCC and D-limonene should be also included in Figure 6 to compare the thermal behavior of the materials.
- Pristine MCC, modified MCC, and also D-limonene has been added. These thermograms were given in another Figure (Figure 6).
- The following statement was made by the authors and which should be proved by some evidence, i.e., SEM of the nanofibers with an excess of D-limonene should be reported.
“Meanwhile, excess D-limonene deteriorated the fiber morphology, caused beaded fiber formation, and reduced the release amount.”
- ''…. Meanwhile, excess D-limonene deteriorated the fiber morphology, caused beaded fiber formation, and reduced the release amount. ÇallıoÄŸlu and Güler (2020) fabricated PVP/gelatin/lavandula essential oil nanofibers [ÇallioÄŸlu, F. C., &Güler, H. K. (2020). Production of essential oil-based composite nanofibers by emulsion electrospinning. PamukkaleÜniversitesiMühendislikBilimleriDergisi, 26(7), 1178-1185.]. They asserted that a certain concentration of lavender causes beads in their nanofibers and their morphological deformation begins. It has also been found that if the amount of Zataria Multiflora (ZM) essential oil is > 10% (v/v), the continuous form of the polymer solution deforms and therefore no nanofibers are produced [Ardekani, N. T., Khorram, M., Zomorodian, K., Yazdanpanah, S., Veisi, H., &Veisi, H. (2019). Evaluation of electrospun poly (vinyl alcohol)-based nanofiber mats incorporated with Zataria multiflora essential oil as potential wound dressing. International journal of biological macromolecules, 125, 743-750.]
- SEM images of fibers after drug-loading and drug-releasing should be reported.
- In this study, the matrix materials (PVA/Psyllium husk) are water-based and the matrix materials are dissolved in water due to the fact that no toxic crosslinkers are used. Therefore, SEM images cannot be obtained. (The meshes are completely dissolved in the release medium.)
- Did the authors do the crosslinking of PVA and its composites fibers? Because without the crosslinking process the fibers immediately dissolved in the aqueous solution/buffer. Therefore, it is difficult to measure drug release.
- In this study, the fibers are not crosslinked. However, the use of beta-cyclodextrin slightly delayed the release of D-limonene. Because of the use of beta-cyclodextrin acts both as a retarding the release of essential oils and as a non-toxic emulsifier in water-based matrixes.
- The authors should report the pH sensitivity of pristine and composite nanofibers. Because they behave differently under a wide range of pHs. Thus, studying the drug release under physiological and other pH solutions is recommended. The authors may follow the following articles as model procedures for pH-responsive systems and quote them in the revised manuscript. https://doi.org/10.1039/D1EN00354B, https://doi.org/10.1016/j.msec.2020.111698, https://doi.org/10.1016/j.msec.2020.110928
- Many thanks for the reviewer's valuable comments. However, the references related to acrylate-based polymeric fibers and therefore acrylate nanofibers are pH-responsive smart systems. In this study, PVA and PSH were used for polymer matrix and wound application systems proposed release analysis to perform in neutral media like PBS (pH 7.2 or 7.4).
(- https://pubmed.ncbi.nlm.nih.gov/34842695/
- https://www.sciencedirect.com/science/article/pii/S2090123218300602
- https://link.springer.com/article/10.1007/s10934-020-01029-1
-https://onlinelibrary.wiley.com/doi/10.1002/cnma.202100349 )
Unfortunately, the proposed references by the respectful reviewer have no relevance to this paper whatsoever, and in order to prevent academic misconduct and respect the ethics committee of the publisher, we are unable to add these references. Regards.
- The mechanical properties of nanofibers are the key characteristics for advanced in vivo drug delivery and tissue engineering applications. The authors should be reported the tensile data in the revised manuscript.
- It is quite difficult to measure mechanical properties in hydrophilic nanofibers. Since the nanofibers were not crosslinked, there were problems when placing them in a tensile test machine. For this reason, this test was also not performed.
- The introduction section is insufficient and should be revised entirely so that the rn identifies the scientific problems solved by this research. Several biocompatible systems have a wide range of biomaterials, including hydrogels, nanoparticles, nanofilms, nanocomposites, etc., but why did the authors select only nanofibers? The authors should emphasize why the nanofibers are familiar, or favor compared to other systems using the following articles. Moreover, the information on biomaterials (BSA, Gelatin, Zein, PCL, PLA, chitosan, UHMWPE, etc.) should be explored in the introduction with recent references, Thus, the following articles should be quoted in the introduction and other sections.
Chitosan (https://doi.org/10.1016/j.colsurfb.2021.111819), BSA (https://doi.org/10.1016/j.msec.2020.111698), Gelatin; (https://doi.org/10.3390/ph14040291, https://doi.org/10.1016/j.jmbbm.2020.103696), Zein (https://doi.org/10.3390/pharmaceutics11120621). https://doi.org/10.1016/j.msec.2020.110928. https://doi.org/10.1039/D1EN00354B, https://doi.org/10.3390/pharmaceutics12121208, https://doi.org/10.1016/j.jmbbm.2021.104554
-We agreed with the reviewer's comments that there are many biocompatible systems available to attract such as hydrogels, nanoparticles, nanofilms, and nanocomposites. The introduction part has been revised and in order to prevent academic misconduct and respect to the ethics committee of the publisher, we added some references are related to the topic. Regards (within red in the introduction part).
A polymeric biomaterial is aimed to interact with biological systems in evaluating, repairing, restoring, or replacing any tissue, organ, or function of the body. [Parın and Parın, 2022] Polymers have a lot of potential because of their chemical flexibility and a wide variety of physical and mechanical characteristics. Polymers have a lot of potential because of their chemical flexibility and a vast variety of physical and mechanical capabilities. Many bio-based polymers, such as gelatin, zein, chitosan, BSA, PCL, and PLA, have been employed in targeted drug delivery systems in various polymeric templates incorporating hydrogel, nanoparticle, nanocomposite, nanofilm, and nanofiber (Mamidi, N., Delgadillo, R. M. V., & González-Ortiz, A. (2021). Engineering of carbon nano-onion bioconjugates for biomedical applications. Materials Science and Engineering: C, 120, 111698. ; Mamidi, N., Delgadillo, R. M. V., & González-Ortiz, A. (2021). Engineering of carbon nano-onion bioconjugates for biomedical applications. Materials Science and Engineering: C, 120, 111698. ; Mamidi, N., Velasco Delgadillo, R. M., & Barrera, E. V. (2021). Covalently functionalized carbon nano-onions integrated gelatin methacryloyl nanocomposite hydrogel containing γ-cyclodextrin as drug carrier for high-performance pH-triggered drug release. Pharmaceuticals, 14(4), 291. ; Mamidi, N., Castrejón, J. V., & González-Ortiz, A. (2020). Rational design and engineering of carbon nano-onions reinforced natural protein nanocomposite hydrogels for biomedical applications. Journal of the Mechanical Behavior of Biomedical Materials, 104, 103696. ; Mamidi, N., González-Ortiz, A., Lopez Romo, I., & V Barrera, E. (2019). Development of functionalized carbon nano-onions reinforced zein protein hydrogel interfaces for controlled drug release. Pharmaceutics, 11(12), 621. ; Mamidi, N., Zuníga, A. E., & Villela-Castrejón, J. (2020). Engineering and evaluation of forcespun functionalized carbon nano-onions reinforced poly (ε-caprolactone) composite nanofibers for pH-responsive drug release. Materials Science and Engineering: C, 112, 110928.)
It would be more realistic to cover such kind of research work in the current manuscript. Which will enrich the quality of the current manuscript as well as inquisitiveness to the readers.

Reviewer 2 Report
The manuscript entitled “Development of PVA/Psyllium husk meshes via emulsion electrospinning: Preparation, characterization, and antibacterial activity” by Parin et al reports an interesting procedure for the incorporation of D-limonene and MCC into the nanofibrous mats to improve the hydrophilicity and the antibacterial properties against gram-negative and gram-positive bacteria strains. The approach sounds interesting for the field of materials science with biomedical applications and the characterization techniques employed are, in general, adequate. However, some aspects need to be solved or clarified. Please, consider the following points to improve your manuscript in order to be accepted for their publication in Polymers:
- Please, when referring to particular bacteria names use the italic letter.
- Abstract. Please, rewrite considering give a brief introduction to the topic previously the current study.
- Introduction. Need to be rewritten in general. Besides, when authors refer to 3D nanofiber meshes, a more deep analysis of the literature should be made to identify the differences between 2D and 3D structures and the corresponding advantages of three-dimensionality. Please, consult the following articles: Materials Science and Engineering: C 110 (2020): 110716, Nanotechnology 31, 17 (2020): 172002, International Journal of Pharmaceutics 610 (2021): 121228.
- Introduction. Improve the paragraphs referred to emulsion electrospinning.
- Introduction. Line 67. Authors said “organic solvents” are highly restricted in electrospinning techniques for biomedical applications. Is a very general statement. I recommend consulting some bibliography: ICH Harmonised Guideline Impurities: guideline for residual solvents Q3C (R6) Current Step, 4 (2016), pp. 1-34, Materials Letters 245 (2019): 86-89, Nanomaterials 6.4 (2016): 75.
- Introduction. Line 75. “C10H16”, rewrite in an adequate form. Same for “cm-1” along the whole manuscript and “-CH2” in the FTIR section.
- Introduction. The end of the introduction should contain a summary of the most important findings of the current work.
- Figures 1 and 2. Please, incorporate conditions and important details in the schemes.
- Why is the reason for heating PVA to dissolve during 3 hours? At 10 % p/V, complete disolution is posible in 30 min.
- Electrospinning procedure. Please, report the temperature and RH %. What about the duration of the process? And the thickness is possible to control?
- Figure 4. Please, improve the presentation of the pictures. Scale bars are not visible. Please, express the distribution diameter plots as relative frequency.
- Figure 5: is “wavenumber”. Please, remark the representative bands in the spectra and incorporate the spectrum of D-limonene.
- Line 234. Carbon dioxide, rewrite.
- Figure 6. TGA for D-limonene is not shown.
- About the cumulative release, why not be expressed as a percentage? By the way, the encapsulation efficiency of the bioactive compound is missing. Please, follow the standard way for informing this kind of results.
- Table 2 is missing… I could only identify Tables 1 and 3.
- If PP0 is only PVA electrospun mat is strange that is not instantaneously dissolved in the contact angle experiment.
- Conclusion. I recommend to the authors incorporate a more deep analysis of potential applications and a future outlook regarding the studied materials. Are the authors planning some assays using infected skin or animal models?
Author Response
Manuscript Number: polymers-1647211
Point to point revision
Manuscript Title: Development of PVA/Psyllium husk meshes via emulsion electrospinning: Preparation, characterization, and antibacterial activity.
Dear Editor,
We thank the reviewers for the evaluations, suggestions and constructive comments on the development of our manuscript. Changes and some additions were suggested by referees in the manuscript have been made. The given answers are presented below one by one.
Reviewer 2
Comments and Suggestions for Authors
The manuscript entitled “Development of PVA/Psyllium husk meshes via emulsion electrospinning: Preparation, characterization, and antibacterial activity” by Parin et al reports an interesting procedure for the incorporation of D-limonene and MCC into the nanofibrous mats to improve the hydrophilicity and the antibacterial properties against gram-negative and gram-positive bacteria strains. The approach sounds interesting for the field of materials science with biomedical applications and the characterization techniques employed are, in general, adequate. However, some aspects need to be solved or clarified. Please, consider the following points to improve your manuscript in order to be accepted for their publication in Polymers:
- Please, when referring to particular bacteria names use the italic letter.
- The special bacteria names have been revised in the manuscript
(In introduction part, '' ……It was determined that these active films showed antimicrobial activity against E. coli, L. monocytogenes, S. aureus, and S. aureus P. aeruginosa bacteria [26], …… These materials also exhibited antibacterial activity against S. aureus and E. coli bacteria.)
(In part 3.6 Antibacterial assays, ''…… Disk diffusion procedure was used the antibacterial activities of D-limonene-loaded meshes and D-limonene-free meshes against three kinds of microbial species (E.coli, S. aureus, and P. aeruginosa) bacteria (Table 3 and Figure 9.) ''. Moreover, the others have been corrected one by one in the manuscript. (Within red)
- Please, rewrite considering give a brief introduction to the topic previously the current study.
- Abstract has been revised again (in red).
- Need to be rewritten in general. Besides, when authors refer to 3D nanofiber meshes, a more deep analysis of the literature should be made to identify the differences between 2D and 3D structures and the corresponding advantages of three-dimensionality. Please, consult the following articles: Materials Science and Engineering: C 110 (2020): 110716, Nanotechnology 31, 17 (2020): 172002, International Journal of Pharmaceutics 610 (2021): 121228.
- The introduction part has been revised in the manuscript (in red). Further, the reference has been cited in the related part in the manuscript (in red). ( Jung, S., Pant, B., Climans, M., Shaw, G. C., Lee, E. J., Kim, N., & Park, M. (2021). Transformation of electrospun Keratin/PVA nanofiber membranes into multilayered 3D Scaffolds: Physiochemical studies and corneal implant applications. International Journal of Pharmaceutics, 610, 121228.)
- Improve the paragraphs referred to emulsion electrospinning.
- The paragraph has been extended.
Green electrospinning has grown in popularity in recent years due to its environmentally friendly, clean, and safe fabrication without toxic materials (Castilla-Casadiego et al., 2016; Güler et al., 2019) As a result of this tendency, the emulsion electrospinning technique, which is the primary way of green electrospinning, has become attractive. The method of preparing polymer solutions is critical to emulsion electrospinning. An emulsion consists of 2 or more immiscible liquid phases, named continuous and disperse phases (Agarwal & Greiner, 2011).
Many studies on nanofibers with essential oils via emulsion electrospinning can be found in the literature (Shin, J., & Lee, S. (2018). Encapsulation of phytoncide in nanofibers by emulsion electrospinning and their antimicrobial assessment. Fibers and Polymers, 19(3), 627-634 ;ÇallioÄŸlu, F. C., &Güler, H. K. (2020). Production of essential oil-based composite nanofibers by emulsion electrospinning. PamukkaleÜniversitesiMühendislikBilimleriDergisi, 26(7), 1178-1185.).Rieger et al. (2016) fabricated chitosan/polyethylene oxide/cinnamaldehyde nanofibers and they observed a relation between solution rheology and the final materials (Rieger, K. A., Birch, N. P., &Schiffman, J. D. (2016). Electrospinning chitosan/poly(ethylene oxide) solutions with essential oils: Correlating solution rheology to nanofiber formation. Carbohydrate Polymers, 139, 131–138. doi:10.1016/j.carbpol.2015.11.073). In another study, cabreuva essential oil loaded into PVA-chitosan nanofibers produvced by Lamarra et al. (2020). (Lamarra, J., Calienni, M. N., Rivero, S., &Pinotti, A. (2020). Electrospun nanofibers of poly (vinyl alcohol) and chitosan-based emulsions functionalized with cabreuva essential oil. International Journal of Biological Macromolecules, 160, 307-318.)In their study, antibacterial activity against micro-organisms (Candida albicans, E. coli, S. aureus, and S. epidermidis) and release behavior of the resulting materials were evaluated. Mouro et al. (2019) developed composite electrospun fibers contains Polycaprolactone (PCL)/Polyvinyl Alcohol (PVA)/Chitosan (CS)/ Eugenol (EUG). They observed resultant nanofibers have a burst release of eugenol from the matrix for the first 8 h. Moreover, antibacterial efficiency of the materials against S. aureus and P. aeruginosa bacteria has been found around 92% and 95%, respectively (Mouro, C., Simões, M., &Gouveia, I. C. (2019). Emulsion electrospun fiber mats of PCL/PVA/chitosan and Eugenol for wound dressing applications. Advances in Polymer Technology, 2019.)
- Line 67. Authors said “organic solvents” are highly restricted in electrospinning techniques for biomedical applications. Is a very general statement. I recommend consulting some bibliography: ICH Harmonised Guideline Impurities: guideline for residual solvents Q3C (R6) Current Step, 4 (2016), pp. 1-34, Materials Letters 245 (2019): 86-89, Nanomaterials 6.4 (2016): 75.
Dear reviewer, we cannot reach your manuscript. Therefore, it cannot be added in the manuscript.
- Line 75. “C10H16”, rewrite in an adequate form. Same for “cm-1” along the whole manuscript and “-CH2” in the FTIR section.
- These terms have been revised again.
- The end of the introduction should contain a summary of the most important findings of the current work.
- The last paragraph has also been revised, again.
There are numerous studies related to psyllium husk gel used in the formulation for making fibers by wet spinning [6,7,43,44]. As far as we know, antibacterial nanofibers consisting of PVA/psyllium husk polyblend are not yet available in the literature. Based on the available literature, in this study, we aimed to develop new totally bio-based and antibacterial electrospun PVA/psyllium husk/D-limonene meshes via emulsion electrospinning. The morphological, chemical, and thermal analyses were examined. Finally, the antibacterial activity of the fibrous meshes against Gram (-) (Escherichia coli, Pseudomonas aeruginosa) and Gram (+) bacteria (Staphylococcus aureus) bacteria was assessed, as well as the release behavior of D-limonene in NaCl media concerning the modified MCC (mMCC) and the varying D-limonene content (2%, 4%, and 6%, v/v).
- Figures 1 and 2. Please, incorporate conditions and important details in the schemes.
- Some details have been added in Figures 1 and 2, and these are revised again (in the manuscript).
- Why is the reason for heating PVA to dissolve during 3 hours? At 10 % p/V, complete disolution is posible in 30 min.
-Dear reviewer, the PVA (ZAG Industrial Chemicals (Istanbul, Turkey)) we use can dissolve it in the range of 80-90 degrees in about 3 hours, unfortunately.
- Electrospinning procedure. Please, report the temperature and RH %. What about the duration of the process? And the thickness is possible to control?
- … The temperature is about 25°C (laboratory condition) and RH is lower than 45% for all the processes. Thicknesses are almost the same (74 – 76 µm.)
- Figure 4. Please, improve the presentation of the pictures. Scale bars are not visible. Please, express the distribution diameter plots as relative frequency.
- DPI of Figure 4 has been improved to 6400.
- Figure 5: is “wavenumber”. Please, remark the representative bands in the spectra and incorporate the spectrum of D-limonene.
- The term has been revised as ''wavenumber''. Moreover, D-limonene and mMCC have been added to the FT-IR spectrum.
- Line 234. Carbon dioxide, rewrite.
- The term has been rewritten.
- Figure 6. TGA for D-limonene is not shown.
- TGA of pure D-limonene, pure MCC and modified MCC has been added.
Characterization Part:
Thermal stability of the nanofibers, MCC, and mMCC was carried out using a TA/SDT650 TGA (USA) in a nitrogen atmosphere with a heating rate of 10 °C min-1 over temperatures ranging from room temperatures to 600 °C, and in an oxygen atmosphere with a heating rate of 10°C min-1 over a temperature range of 600–900 °C. Similarly, TGA analysis of D-limonene was applied in a nitrogen atmosphere with a heating rate of 10 °C min-1 over temperatures ranging from room temperatures to 600 °C.
Results and Discussion Part:
Thermal stability of D-limonene is shown in Figure 6. Based on the results, free D-limonene evaporated with ranging from 50°C to 135°C. The residue was 1.25%. Levic et al. (2011) confirmed the same thermal curve of D-limonene in N2 atmosphere. The majority of limonene evaporated at a somewhat lower temperature range than that reported by Levicet al. Variations in the thermal characteristics of D-limonene can be caused by variation in experimental setup or by the chemical composition of D-limonene.
The initial weight losses for pure and modified cellulosic structures beginning at 100°C can be explained by the evaporation of moisture on the surfaces of MCC and mMCC samples. The TGA curves of these structures exhibit a similar degradation trend. The first step of MCC and CMC degradation occurs at ranging from 285-370°C and 250-360 °C, respectively, with a total weight loss of 87% and 85%, respectively. Cellulose degradation occurs at a max temperature of around 350 °C due to the scissoring of glycosidic linkage in cellulose was found by Kuthi et al. (2016). Furthermore, a greater start temperature is linked with high thermal stability (Nasutin et al. 2017). The hydrolysis process (Figure 1) has made the mMCC more susceptible to degradation as the temperature increases (Figures 6 a and b). As the increase of the hydrolytic ratio, the molecular weight and Tmaxvalues decrease. This is due to the crystalline fraction of the cellulose's hydrolytic activity, as well as amorphous hydrolysis (Kuthi et al. (2016).
Kuthi, F. A. A., Norzali, N. R. A. A., &Badri, K. H. (2016). Thermal Characteristics of microcrystalline cellulose from oil palm biomass. Malays. J. Anal. Sci, 20, 1112-1122.
Nasution, H., &Sitompul, S. (2017, March). Preparation and characterization of cellulose microcrystalline (MCC) from fiber of empty fruit bunch palm oil. In IOP Conference Series: Materials Science and Engineering (Vol. 180, No. 1, p. 012007). IOP Publishing.
Figure 6. TGA and DTG thermograms of pristine MCC, modified MCC, and D-limonene.
- About the cumulative release, why not be expressed as a percentage? By the way, the encapsulation efficiency of the bioactive compound is missing. Please, follow the standard way for informing this kind of results.
Cumulative release (%) = (1)
where Mt is the total amount of D-limonene released at usual time intervals t and M0 is the beginning amount of D-limonene in the electrospun meshes.
Encapsulation efficiency (%) = x 100
Due to the dissolution of the entire meshes in the release medium, the encapsulation efficiency is also equal to the maximum amount of release.
- Table 2 is missing… I could only identify Tables 1 and 3.
- Dear editor, thank you for your attention. Table 3 has been changed to Table 2.
- If PP0 is only PVA electrospun mat is strange that is not instantaneously dissolved in the contact angle experiment.
- PP0 sample is PVA electrospun meshes. As you see in the Figure 8, the meshes have the lowest contact angle value (lower than 15°). Actually, it can be described PVA meshes are not directly dissolved in liquid media, as is the case with UV-Vis release (in-vitro release) analysis. Firstly, it showed are not directly dissolved swelling behavior by absorbing liquid into its structure and then it was broken down and dissolved.
- I recommend to the authors incorporate a more deep analysis of potential applications and a future outlook regarding the studied materials. Are the authors planning some assays using infected skin or animal models?
- Conclusion part has been extended (in red).
PVA/PSH electrospun meshes containing both D-limonene and mMCC were efficiently produced by emulsion electrospinning, and D-limonene was used to model an antibacterial agent. MCC was treated with a silane coupling agent which is known (3-chloropropyl) triethoxysilane (CPTES) to obtain mMCC. The microstructural analysis indicated that an excessive increase in the amount of D-limonene caused the beaded fiber formation. Moreover, the average fiber diameter of meshes decreased from 456.53 ± 56.9 nm (for PP1) to 274.22 ± 68.25 nm (PPMD6) due to the increase in D-limonene. In FTIR spectroscopy, the characteristic peaks of the matrix overlapped the distinct peaks of D-limonene. The characteristic peaks of D-limonene were appointed slightly in the 1000-1200 cm-1 band. Further, only the addition of MCC increased the thermal stability based on thermal analysis. It was reported the increase of D-limonene increases the contact angle, but the contact angle of all D-limonene-containing samples may be lower due to β-cyclodextrin compared to the neat sample (PP1). Modified MCC to polyblend meshes increased both D-limonene release and antibacterial activity of resulting meshes. According to the findings, D-limonene revealed more antibacterial activity against gram-negative bacteria (E.coli) than gram-positive bacteria (S. aeurus) highlighting that electrospun meshes are suitable for carrying D-limonene and effectively increasing its antimicrobial activity. It supports the evidence that D-limonene continues in its therapeutically active form once loaded into electrospun fibers. We hope that our results will encourage people to recognize the importance of traditional natural molecules like D-limonene for their therapeutic potential advantages, as well as develop the pharmaceutical, and cosmetic fields for efficient protection and release of such natural molecules. Future research should investigate at the release behavior and associated activities of bioactive agents (combining another antibacterial molecule with D-limonene) when loaded in electrospun fibers made from diverse polymer combinations (PSH and another matrix). The formulations are likely to be more successful when tested in appropriate in-vivo models. In summary, the study demonstrated that D-limonene might support as an inhibitor of bacteria. Therefore, the obtained new polyblend meshes can be candidates for wound healing applications.

Round 2
Reviewer 1 Report
Suggested to accept the revised manuscript in its current form.